# Capsular Polysaccharide Is a Receptor of a *Clostridium perfringens* Bacteriophage CPS1

**DOI:** 10.3390/v11111002

**Published:** 2019-10-31

**Authors:** Eunsu Ha, Jihwan Chun, Minsik Kim, Sangryeol Ryu

**Affiliations:** 1Department of Food and Animal Biotechnology, Department of Agricultural Biotechnology, and Research Institute of Agriculture and Life Sciences, Seoul National University, Seoul 08826, Korea; esha0521@gmail.com (E.H.); jihwanchun@gmail.com (J.C.); 2Department of Food and Nutrition, College of Human Ecology, Yonsei University, Seoul 03722, Korea

**Keywords:** bacteriophage, *Clostridium perfringens*, phage receptor, capsular polysaccharides

## Abstract

*Clostridium perfringens* is a Gram-positive, anaerobic, and spore forming bacterium that is widely distributed in the environment and one of the most common causes of foodborne illnesses. Bacteriophages are regarded as one of the most promising alternatives to antibiotics in controlling antibiotic-resistant pathogenic bacteria. Here we isolated a virulent *C. perfringens* phage, CPS1, and analysis of its whole genome and morphology revealed a small genome (19 kbps) and a short noncontractile tail, suggesting that CPS1 can be classified as a member of *Picovirinae*, a subfamily of *Podoviridae*. To determine the host receptor of CPS1, the EZ-Tn5 random transposon mutant library of *C. perfringens* ATCC 13124 was constructed and screened for resistance to CPS1 infection. Analysis of the CPS1-resistant mutants revealed that the *CPF_0486* was disrupted by Tn5. The *CPF_0486* was annotated as *galE*, a gene encoding UDP-glucose 4-epimerase (GalE). However, biochemical analyses demonstrated that the encoded protein possessed dual activities of GalE and UDP-*N*-acetylglucosamine 4-epimerase (Gne). We found that the *CPF_0486*::Tn5 mutant produced a reduced amount of capsular polysaccharides (CPS) compared with the wild type. We also discovered that glucosamine and galactosamine could competitively inhibit host adsorption of CPS1. These results suggest that CPS acts as a receptor for this phage.

## 1. Introduction

*Clostridium perfringens* is a Gram-positive, spore-forming, anaerobic bacterium that is widely distributed in nature [1]. *C. perfringens* spores can persist in the environment, and the bacteria cause infectious diseases, such as gas gangrene and necrotic enteritis, in humans and animals. Enterotoxigenic strains of *C. perfringens* can cause food poisoning [2,3]. According to the Centers for Disease Control and Prevention (CDC), *C. perfringens* is ranked as one of the five most common food-borne pathogens in the United States [4].

Bacteriophages (or phages) are defined as viruses that specifically infect and kill bacteria. Interest about phage therapy has recently been revived because of the worldwide problem of antibiotic-resistant bacteria [5]. The host range of phage is narrow, so phages can be used to destroy target bacteria without disrupting the natural microflora. A phage cocktail composed of multiple phages is used to increase the phage host range [6,7] and to delay the appearance of phage-resistant bacteria [8]. Phage infection occurs when the phage tail recognizes receptors on host bacteria, and infection is followed by phage DNA injection and eventual by phage replication. Bacteria can develop phage resistance by modification or masking of the receptors [9,10]. If the phage cocktail is made with various phages utilizing different receptors, this phenomenon can delay the development of resistant bacteria, since it is much more difficult for bacteria to mutate a number of different receptors [11,12]. These findings suggest that identification of the phage receptor is essential to improve the efficiency of phage therapy [13,14].

Although several phage receptors of Gram-positive bacteria, such as cell wall teichoic acids (WTA), cell wall-associated polysaccharides, and lipoteichoic acids (LTA), have been reported [10,15], relatively little is known about the host receptors of phages infecting Gram-positive bacteria compared with those of phages infecting Gram-negative bacteria [16]. It is known that a complex of peptidoglycan and WTA can competitively inhibit phage adsorption to *Staphylococcus aureus* [17] and that glycosylation of WTA can prevent *S. aureus* phage infection [17,18]. *Bacillus* phage SPP1, SP10, and SP02 are unable to infect *Bacillus* mutants lacking WTA [16,17]. *Lactobacillus* phages are inactivated when exposed to LTA isolated from the host cell wall [10,17]. A few *C. perfringens* phages have been reported [19,20,21]; however, none of their host receptors have been identified.

In this study, we isolated the *C. perfringens* virulent bacteriophage CPS1. The phylogenetic and morphological analysis categorized CPS1 as a member of *Picovirinae*, a *Podoviridae* subfamily. Screening of a random transposon mutant library of *C. perfringens* ATCC 13124 revealed that *C. perfringens* capsular polysaccharides (CPS), synthesized by bifunctional UDP-galactose (UDP-Gal)/UDP-*N*-acetylgalactosamine (UDP-GalNAc) 4-epimerase, served as a CPS1 phage receptor. This is the first report to identify the host receptor of *C. perfringens* phage.

## 2. Materials and Methods

### 2.1. Bacteria and Bacteriophage Growth Conditions

The *C. perfringens* strains used in this research, ATCC 13124, were acquired from Korean Collection for Type Cultures (KCTC). The bacterial cells were grown in brain heart infusion (BHI) medium with appropriate antibiotic supplements (chloramphenicol 5 μg/mL, erythromycin 10 μg/mL) at 30 °C under anaerobic conditions. CPS1 was isolated from Livestock Waste Treatment Plants in Namyangju, Korea. Phage isolation method was previously described [20].

### 2.2. Morphological Analysis by TEM

Purified CPS1 (1 × 10^9^ PFU/mL) was placed on carbon-coated copper grids and negatively stained with 2% aqueous uranyl acetate (pH 4.0) for 20 s. The morphology of CPS1 was analyzed by transmission electron microscopy (TEM, LEO 912AB transmission electron microscope; Carl Zeiss, Wezlar, Germany). TEM images were scanned at the National Academy of Agricultural Science (Jeonju, Korea).

### 2.3. DNA Purification and Whole-Genome Sequencing of Bacteriophage CPS1

To extract genomic DNA from CPS1, host DNA was removed by treatment of the virions with DNaseI and RNaseA (1 μg/mL each) at room temperature for 30 min. The virions were then lysed by incubating them with a mixture (50 μg/mL proteinase K, 20 mM ethylenediaminetetraacetic acid (EDTA), 0.5% sodium dodecyl sulfate (SDS)) at 56 °C for 1 h. After lysis, the DNA was purified by phenol-chloroform extraction and precipitated by ethanol [22,23]. The purified genomic DNA of CPS1 was sequenced using the GS-FLX Titanium sequencer (Roche Holding AG, Basel, Switzerland). A total of 1225 sequencing reads obtained were assembled using GS De Novo Assembler version 2.9 (Roche Holding AG, Basel, Switzerland) with default parameters. Additional DNA sequencing of the phage was performed to identify end regions of the genomic DNA sequence using primers in Macrogen Inc. (Seoul, South Korea) (Appendix A). The position of open reading frames (ORFs) was predicted using bioinformatics tools, including Glimmer 3.02 and Rapid Annotation using Subsystem Technology (RAST) software [24,25]. The function of each ORF was predicted using NCBI BLASTP and InterProScan databases [24]. Based on the information obtained, the name of each ORF was annotated manually. The complete genome sequence of CPS1 was deposited in GenBank under accession number KY996523.

### 2.4. Phylogenetic Analysis

To determine the phylogeny of CPS1 and the ATCC 13124 CPF_0486 protein, we plotted phylogenetic trees based on the alignment of the amino acid sequences from the DNA polymerase of various bacteriophages or NAD^+^-dependent epimerase of various organisms. The amino acid sequences used for phylogenetic trees are available online in the NCBI nucleotide databases (http://www.ncbi.nlm.nih.gov/nuccore). The amino acid sequences were aligned using the Clustal X2 program [26]. The phylogenetic trees were constructed with MEGA7 by the neighbor-joining method and bootstrap analysis (2000 replicates) with *P*-distance values [27,28].

### 2.5. Molecular Genetic Method

The *C. perfringens* ATCC 13124 *CPF_0486*::Tn5 mutant was isolated from EZ-Tn5 mutagenesis library. A random *C. perfringens* ATCC 13124 mutant pool was generated using the EZ-Tn5 kit as described by the manufacturer (Epicentre^®^, Madison, WI, USA). In brief, the electrocompetent cells were prepared as described previously [29] and 400 μL of electrocompetent cell was mixed with the EZ-Tn5 transposome plus short-fragment DNA (sfDNA, 300 bps) and electroporated using a Bio-Rad Gene Pulser^TM^ with pulse controller set at 200 Ω, 25 μF, 1.5 kV and 0.2-cm cuvette width [29,30].

The efficiency of Erm-transposon transformation was ~2500 transformants/reaction. This mutant pool was mixed with CPS1 phage by the high-titer-overlay method (10^9^ PFU/mL) on a BHI agar plate with erythromycin. Phage-resistant mutants appeared after 30 h of incubation at 30 °C. When phage-resistant colonies were developed, gDNA was purified from the colonies and sequenced using the appropriate primers listed in Appendix A. The resulting sequence data were compared with the ATCC 13124 gDNA sequence by BLAST analysis.

For complementation, the *E. coli*-*C. perfringens* shuttle vector pJIR750 was used (Appendix A). *CPF_0486* was amplified containing the upstream 403 bp and the downstream 281 bp. pJIR750 was digested with *BamH1* and *Sal1*. ATCC 13124 *CPF_0486*::Tn5 was transformed with pJIR750::*CPF_0486*. To confirm complementation, the bacteriophage dotting assay was performed with CPS1.

### 2.6. ATCC 13124 CPF_0486 Expression and Purification

The *CPF_0486* gene from the *C. perfringens* ATCC 13124 genome was amplified using primers (Appendix A) containing either a terminal *EcoRI* or *BamHI* site to introduce appropriate restriction enzyme sites for subcloning into pET28a. The PCR product was purified, digested with *EcoRI* and *BamHI*, and purified using the Wizard DNA purification kit (Promega, Madison, WI, USA), and introduced into similarly digested, purified pET28a via conventional means. The constructed N-terminally His_6_-tagged CPF_0486 expression vector was transformed into *E. coli* BL21 (DE3).

The gene encoding CPF_0486 with an N-terminal His_6_-tag was expressed in *Escherichia coli* strain BL21 (DE3) and purified by Ni-agarose affinity chromatography. To express the protein, the precultured transformants were inoculated in LB broth containing 50 μg/mL kanamycin to an OD_600_ of 0.7 and induced with 0.5 mM isopropyl-β-d-thiogalactopyranoside (IPTG). Then, the induced culture was incubated for 12 h at 30 °C, 200 rpm. The *E. coli* cells were harvested by centrifugation (10,000× *g* for 10 min at 4 °C), and the cell pellet was resuspended in lysis buffer (100 mM NaCl, 50 mM Tris-HCl) and disrupted by sonication (Branson Ultrasonics, Danbury, CT, USA). After centrifugation at 16,000 × *g* for 40 min, the supernatant was purified using a Ni-nitrilotriacetic acid (NTA) Superflow column (Qiagen, Hilden, Germany) according to the manufacturer’s instructions. The identity and purity of the protein were assessed by sodium dodecyl sulfate polyacrylamide gel electrophoresis (SDS-PAGE). The purified CPF_0486 protein was stored at 80 °C after the buffer was replaced with storage buffer (4 mM NAD^+^, 50% glycerol) using PD Miditrap G-25 (GE Healthcare, Chicago, IL, USA) [31]. To remove the His_6_ tag from CPF_0486 protein, we used the thrombin kit (Novagen, Madison, WI, USA). The enzyme assays were performed after removal of the His_6_-tag with thrombin.

### 2.7. ATCC 13124 CPF_0486 Protein Characterization

The UDP-galactose 4-epimerase activity was measured with the colorimetric assay as described by Fry et al. [32]. The reaction was conducted at 37 °C for 10 min in a total volume of 0.5 mL (pH 9.0), which contained 10 mM glycine, 1 mM MgCl_2_, 0.2 mM UDP-Gal and 10 μg of CPF_0486 protein [32,33]. After 10 min, the reaction was halted by adding 1 μL of 10 M HCl and boiling for 5 min. To neutralize the mixture, 1.25 μL of 10 M NaOH was added. 4-epimerase activity with glucose was assayed by adding 2 mL of 0.1 M phosphate buffer (pH 7.0) containing 200 μg of glucose oxidase, 10 μg of peroxidase, and 600 μg of o-dianisidine. After 30 min, the reaction was stopped by adding 2.0 mL of 100% H_2_SO_4_, and the color was measured at 540 nm.

The UDP-*N*-acetylgalactosamine 4-epimerase activity assay has been previously described [33]. The enzyme assay was performed identically to the assay for UDP-GalNAc with the following assay components per 0.5 mL: 10 mM glycine, 1 mM MgCl_2_, 0.2 mM UDP-GalNAc, and 10 μg of CPF_0486 protein. The reaction was performed at 37 °C for 10 min and stopped by addition of 1 μL of 10 M HCl and boiling for 30 min. After neutralization with 1.25 μL of 10 M NaOH, 50 μL of freshly prepared 1.5% (*v*/*v*) acetic anhydride in acetone was added. After 5 min at room temperature, 150 μL of a 0.7 M potassium tetraborate solution was added, and the mixture was immediately boiled for 3 min. After cooling, 300 μL of DMAB reagent was added without shaking, followed by the addition of 2.7 mL of glacial acetic acid [34]. After incubation at 37 °C for 30 min, the color was measured at 585 nm.

### 2.8. In Vitro Bacteriophage Adsorption Assays

Bacterial cells (50 mL; OD_600_ = 1.0) were harvested by centrifugation (7000× *g*; 10 min; 4 °C) and resuspended in 50 mL fresh BHI broth. The suspension of bacterial cells was diluted at an OD_600_ of 0.1 (approximately 10^7^ CFU/mL) in BHI broth, and the suspension was aliquoted into 15-mL Falcon tubes (10 mL suspension per tube). The cells were supplemented with 25 μg/mL of chloramphenicol to prevent cell growth and phage multiplication [10]. Then, 100 μL of 10^7^ PFU/mL CPS1 was added to each tube and incubated at 30 °C. Samples were collected at 5-min intervals, and the cells were immediately removed by centrifugation (16,000× *g*; 1 min; 4 °C) and filtration (0.22-μm-pore-size filter). The amount of unbound free phage particles was determined by the phage dotting assay.

For the periodate and proteinase K treatment adsorption assay, we followed a previously described method [10,35]. Briefly, exponentially growing *C. perfringens* ATCC 13124 cells were treated with proteinase K (0.2 mg/mL; Qiagen) at 30 °C for 2 h. For the periodate treatment group, the cells were centrifuged, and the pellet was suspended in sodium acetate (50 mM; pH 5.2) or sodium acetate containing either 10 or 100 mM periodate (Sigma, St. Louis, MO, USA) before incubation for 2 h in the dark. Following incubation, the samples were washed twice with fresh BHI broth. Adsorption assays were conducted for the various treated cells as described in the results section.

### 2.9. Capsule Staining

Capsules were negatively stained using Anthony’s capsule stain [36]. Briefly, 1 mL of stationary-phase *C. perfringens* culture was harvested by centrifugation and resuspended in 1 mL of carnation skim milk (skim milk powder resuspended in distilled and deionized water at 1% *w*/*v*). A loop of each resuspended culture was spread onto a clean microscope slide and allowed to dry in the air. Smears were then stained with 1% crystal violet for 5 min and washed gently and thoroughly with 20% copper sulfate. Then, the slides were allowed to dry in the air and subsequently examined using oil-immersion phase contrast microscopy at 100× magnification.

### 2.10. Quantification of Capsular Polysaccharides

*C. perfringens* ATCC 13124 harboring pJIR750, *C. perfringens* ATCC 13124 *CPF_0486*::Tn5 harboring pJIR750 and *C. perfringens* ATCC 13124 *CPF_0486*::Tn5 harboring p*CPF_0486* were treated with appropriate selective antibiotics. Capsular polysaccharides were quantified according to a previously described method [37]. Briefly, bacterial cells were grown to stationary phase in BHI broth with antibiotics. Each normalized culture was harvested by centrifugation (10,000× *g*; 10 min; 4 °C) and washed with PBS. The pellets (0.7 g) were suspended in 10 mL of sodium phosphate buffer (50 mM, pH 7.0) containing 70 mg lysozyme (Sigma). After incubation at 37 °C for 24 h, the supernatant was recovered following centrifugation. The supernatant was treated with 10 mg of DNase I (Roche) and 5 mg of RNase A (Roche) at 37 °C for 1 h. Proteinase K (10 mg) was added for 2 h at 55 °C. The supernatant was adjusted to a concentration of 30% (*v*/*v*) cooled ethanol and allowed to stand at 4 °C for 2 h. The precipitate formed was removed by centrifugation, and the supernatant was adjusted to a concentration of 80% cooled ethanol and precipitated. The precipitate was recovered by centrifugation and dried completely. The pellet was resuspended in 3 mL of distilled and deionized water. We used a modified phenol-sulfuric acid method. Phenol-sulfuric acid colorimetric carbohydrate quantification was conducted in the following manner: 600 μL of the sample and 300 μL of 5% phenol were mixed together in a glass test tube; 1.5 mL of 93% sulfuric acid was added, and the tube was agitated to ensure mixing and then allowed to develop color for 10 min. Eventually, the intensity of color was measured at 490 nm in a spectrophotometer. The standard curve of the trisaccharide mixture (rhamnose, galactose, and *N*-acetylgalactosamine; molecular ratio, 1:1:1) was generated as above described, and concentration of the extracted CPS were interpolated.

### 2.11. Inhibition of Phage Infection by CPS or Monosaccharide

Inactivation of CPS1 was examined as previously described with modifications [38]. Briefly, dilutions of phage CPS1 (10^5^ PFU/mL) were mixed with equal volumes of CPS extracted from each strain (100 μg/mL) or monosaccharides (400 mM or 40 mM) and incubated at 25 °C for 1 h. The mixtures were diluted 10-fold with SM buffer, and the number of remaining active phages was counted using the double-agar layer overlay assay. The number of phages from a parallel assay conducted with 10% SM buffer was used as a negative control.

### 2.12. Statistical Analysis

Statistical analysis was performed using GraphPad Prism (GraphPad Software, Inc., La Jolla, CA, USA, Version 5.01). Statistically significant differences were calculated by one-way ANOVA with Tukey’s multiple comparison test are indicated. *P* values < 0.05 were considered significant.

## 3. Results

### 3.1. C. perfringens Bacteriophage CPS1 Is a Member of the Picovirinae

*C. perfringens* bacteriophage CPS1 was isolated from animal feces samples collected in Namyangju, Korea, and *C. perfringens* ATCC 13124 was used as a host. Phage CPS1 forms clear plaques on several strains of tested *C. perfringens* (Appendix A). Transmission electron microscopy (TEM) images of CPS1 displayed a small icosahedral capsid of approximately 40 nm in diameter and a short noncontractile tail (Figure 1A).

The CPS1 genome was sequenced using the GS-FLX next-generation sequencing platform (Roche) with 73.64x coverage. The genome consisted of a linear dsDNA of 19,089 base-pairs with 25 open reading frames (ORFs) (Figure 1B). No genes homologous to known genes associated with lysogeny were identified. The 166-bp inverted terminal repeat (ITR) regions at both ends were identified as in other *Picovirinae* phages (GenBank accession number KY996523) [39]. BLASTP analysis of the CPS1 genome revealed that ORF_6 encoded a putative type-B DNA polymerase, which requires terminal proteins to prime DNA replication [39]. We aligned the sequences of *Picovirinae* phage type-B DNA polymerase and compared the phylogenetic relationship of the family [39]. The results classified CPS1 as a member of *Picovirinae* (Figure 1C).

### 3.2. CPS1 Binds to C. perfringens Cell Surface Polysaccharides

The phage adsorption assay revealed that approximately 95% of CPS1 was bound to *C. perfringens* after 40 min of incubation (Figure 2A). To clarify whether a receptor for CPS1 is a cell surface protein or polysaccharide, a CPS1 adsorption assay was performed after periodate or proteinase K treatment of *C. perfringens* cells [13,40]. As periodate degrades saccharide rings with vicinal diols [35], treatment of periodate can disrupt cell surface polysaccharides, whereas proteinase K can digest cell surface proteins. The assay indicated that the treatment of periodate reduced CPS1 adsorption by *C. perfringens*, but proteinase K did not affect the binding of CPS1 to the cell surface (Figure 2B). These results suggest that cell surface polysaccharides play a crucial role in CPS1 infection.

### 3.3. Screening of a Random Mutant Library of C. perfringens for CPS1 Resistance

We attempted to identify a CPS1 receptor gene using an EZ-Tn5 random transposon mutagenesis library of *C. perfringens* ATCC 13124 [41]. Unfortunately, electroporation was extremely difficult due to the presence of the restriction modification system in *C. perfringens* ATCC 13124 [28]. Thus, short-fragment DNA (sfDNA) was tested as a competitor of nuclease during transposon electroporation [30]. The efficiency of the electroporation method with sfDNA is known to be dose and length-dependent [30]. sfDNA molecules of different lengths (50, 300, 600, or 1000 bps) were tested, and the 300-bps sfDNA provided the best results for increasing the transformation efficiency (Appendix A). The Tn5 random mutant library containing 2500 mutants was infected with CPS1 phage, and the CPS1-resistant mutants were isolated. Seven colonies were obtained from the library, and the Tn5 insertion sites were sequenced, revealing that *CPF_0486*, *CPF_0596*, and rRNA genes were disrupted by Tn5 insertion. We chose a *CPF_0486*::Tn5 mutant for further study considering that this gene is in a polysaccharide biosynthesis cluster (Figure 3A) [42] and the periodate treatment decreased CPS1 adsorption to *C. perfringens* as described above. The sensitivity of a *CPF_0486*::Tn5 mutant against CPS1 was restored by complementation (Figure 3B). Comparison of phage adsorption between the wild type and the *CPF_0486*::Tn5 mutant showed that CPS1 did not bind to the *CPF_0486*::Tn5 mutant (Figure 3C).

### 3.4. CPF_0486 Has UDP-glucose 4-epimerase (GalE) and UDP-N-acetylglucosamine 4-epimerase (Gne) Activities

*CPF_0486* was annotated as *galE*, the gene encoding UDP-glucose (UDP-Glc) 4-epimerase or UDP-Gal 4-epimerase [28]. In previous studies, *galE* and *gne* were found to encode highly homologous proteins in *Vibrio vulnificus* and *Bacillus subtilis* (Figure 4A) [43,44]. These reports led us to investigate the characteristics of the CPF_0486 product to verify whether it exhibits UDP-GalNAc 4-epimerase activity [45].

First, we tried to confirm GalE activity in the *CPF_0486* gene product. A glucose oxidation (GO) assay was conducted to examine the potential GalE activity of the CPF_0486 product [32]. The protein catalyzed the conversion of UDP-Gal into UDP-Glc (Figure 4B). Next, we evaluated the Gne activity of the product. The Morgan–Elson reaction assay was used to investigate the UDP-GalNAc 4-epimerase activity [46]. The CPF_0486 product epimerized UDP-GalNAc to UDP-GlcNAc (Figure 4C). These findings indicate that CPF_0486 of *C. perfringens* ATCC 13124 may function as both a UDP-Gal 4-epimerase and a UDP-GalNAc 4-epimerase.

### 3.5. Capsular Polysaccharides are a Receptor for C. perfringens Phage

Gne has been reported to be associated with lipopolysaccharide biosynthesis in *Vibrio vulnificus* [43]. In another study, Gal, Glc, GalNAc, and GlcNAc were shown to be the components of *Streptococcus suis* capsular polysaccharides [47]. Additionally, according to the annotation of ATCC 13124, *CPF_0486* clustered with several genes predicted to be involved in putative capsular polysaccharides (CPS) synthesis [28]. These reports suggest that the gene would be involved in the synthesis of CPS, indicating that the *CPF_0486*::Tn5 mutant might have an impaired ability to produce capsule. Anthony’s capsule staining method was adopted to visualize CPS of wild type and the *CPF_0486*::Tn5 mutant [48]. We found that wild type strain exhibited a halo around the cells, indicative of the presence of a capsule. The halo was greatly reduced in the *CPF_0486*::Tn5 mutant, indicating reduced capsule production. Capsule synthesis was recovered by complementation of the *CPF_0486*::Tn5 mutant (Figure 5A).

We quantified the CPS for each strain. The CPS were extracted and quantified from stationary-phase cultures of each strain. The *CPF_0486*::Tn5 mutant strain produced approximately half the amount of CPS compared with the wild type strain, and CP production was restored in the complemented strain (Figure 5B). These results indicate that *CPF_0486* is associated with CP synthesis in *C. perfringens*. In addition, the specificity of CPS1 binding to CPS was examined by comparing the ability of the CPS extracted from different strains of *C. perfringens* to inhibit CPS1 infection of *C. perfringens* ATCC 13124. The efficiency of plating (E.O.P.) of CPS1 on *C. perfringens* ATCC 13124 was reduced when CPS extracted from *C. perfringens* ATCC 13124 were mixed with the bacteria (Figure 5C). However, CPS extracted from C. *perfringens* FD1, which is not a CPS1 host, were unable to interfere with CPS1 infection (Figure 5C). These results indicate that CPS are a receptor of CPS1.

The repeating structure of trisaccharide (rhamnose, Gal, and GalNAc) units in *C. perfringens* ATCC 13124 capsular polysaccharides (CPS) was recently reported [49]. Based on the idea that CPS are the key components of the CPS1 receptor, we tested the effect of monosaccharides comprising CPS on CPS1 infection. The E.O.P. of CPS1 pre-incubated with monosaccharides, such as Glc, Gal, glucosamine, galactosamine, GlcNAc, GalNAc, and rhamnose, was measured. While GalNAc, GlcNAc, Glc, Gal, and rhamnose did not exhibit any effect on E.O.P., glucosamine and galactosamine inhibited CPS1 infection (Figure 6). These results also support the conclusion that CPS are the phage receptor for *C. perfringens*.

## 4. Discussion

Using *C. perfringens* ATCC 13124 as the host bacterium, we isolated *C. perfringens*-specific bacteriophage CPS1. Morphological analysis by TEM revealed that CPS1 has an icosahedral head and a short noncontractile tail. Structurally, CPS1 can be considered a member of the order *Caudovirales* in the family *Podoviridae* [39]. The CPS1 genome consists of a linear dsDNA strand of 19,089 base-pairs. The ITRs of the CPS1 genome are 166 nucleotide pairs in length, which is much longer than the ITRs of other *C. perfringens* phages belonging to the subfamily *Picovirinae* [39], although only 3 such phages have been identified to date. Additionally, CPS1 contains type-B DNA polymerase requiring a terminal protein to prime DNA synthesis. Phylogenetic analysis of the polymerase amino acid sequences showed that the polymerase of CPS1 is closely related to other *C. perfringens Picovirinae* phages. These genomic properties suggest that CPS1 is a member of the *Picovirinae* subfamily.

Phages encounter the bacterial cell surface randomly by Brownian motion, dispersion, diffusion or flow [16]. Phages bind to bacterial receptors during such processes. After attachment to a receptor, phage molecules undergo conformational changes to insert their genomic DNA into the host cytoplasm. According to previous studies, most Gram-positive bacteria-targeting phages bind to WTA or LTA on the cell surface [16]. Even though CPS are found in both Gram-positive and Gram-negative bacteria, CPS have been reported as a phage receptor only in Gram-negative bacteria [9,16]. However, we found that *C. perfringens* phage CPS1 used CPS as a receptor in this study, which is, to our knowledge, the first report of a phage infecting Gram-positive bacteria via CPS as a receptor [16].

The EZ-Tn5 random mutagenesis system was used to identify a receptor of *C. perfringens* phage. Unlike *C. perfringens* strains 13 and SM101, ATCC 13124 is known for its associated difficulty in genetic transformation by electroporation [28]. Although we improved the electroporation efficiency of ATCC 13124 using sfDNA, mutant generation using the EZ-Tn5 transposon was still low in ATCC 13124, but phage-resistant mutants were isolated. Among the screened EZ-Tn5 *C. perfringens* phage-resistant mutants, 5 out of 7 mutants carried a transposon inserted in the rRNA genes. In a previous study, the EZ-Tn5 transposon has been reported to exhibit higher insertion rates in *C. perfringens* rRNA genes than expected [41]. In other mutants, transposons were inserted in *CPF_0596* or *CPF_0486*, which were annotated as group 2 family of glycosyl transferase and *galE*, respectively. We selected the *CPF_0486* gene for this study because substrates of *galE* can be specified based on genetic analysis.

In previous studies, enzymatic activities of the protein encoded by *gne* have been assessed in *Vibrio vulnificus*, *Campylobacter jejuni*, *Aeromonas hydrophila*, and *Bacillus subtilis*. The product of *gne* exhibited activities of both UDP-Glc 4-epimerase and UDP-GlcNAc 4-epimerase in those species, whereas the *galE*-encoded protein had UDP-Glc 4-epimerase activity only [43,44]. Although *C. perfringens* ATCC 13124 does not carry any gene annotated as *gne*, two ORFs, *CPF_0282* and *CPF_0486,* were annotated as *galE* [28]. In this study we showed that CPF_0486 was responsible for the conversion of UDP-Gal to UDP-Glc as well as UDP-GalNAc to UDP-GlcNAc, suggesting that the *CPF_0486* gene annotated as *galE* may actually be *gne*. We supposed that CPF_0486 can also catalyze the reverse reaction because typical nucleotide sugar epimerases are known to catalyze reversible reactions [43,44]. Moreover, we hypothesized that the *CPF_0486* gene would be involved in the biosynthesis of CPS based on genetic analysis. According to the genome report of *C. perfringens* ATCC 13124, genes associated with the L-rhamnose pathway cluster were predicted to be located upstream of *CPF_0486* [28]. CPS in *C. perfringens* ATCC 13124 are composed of repeating units of trisaccharide (rhamnose, Gal and GalNAc) [49]. The *CPF_0486*::Tn5 mutant constructed in this research was expected to have no CPS because it lacks GalNAc due to *CPF_0486* deletion. In contrast to our expectations, the mutant could still produce a reduced amount of CPS (Figure 5A,B). Bioinformatic analysis of *C. perfringens* ATCC 13124 genome revealed that the *CFP_0282* gene, also annotated as *galE*, may have similar function as the *CFP_0486* gene. Since UDP-Glc and UDP-GlcNAc are common precursors of CPS in the bacterial cell, *CPF_0282* may have contributed to the production of CPS in the *CPF_0486* mutant. A follow-up study is required to verify this assumption.

We expected that GalNAc would inhibit CPS1 infection because CPS of wild type competitively inhibited CPS1 infection (Figure 5C). However, infection of CPS1 was inhibited by glucosamine and galactosamine rather than GalNAc (Figure 6). Similar phenomena have been previously reported [50,51]. In those cases, glucosamine residues were fully *N*-acetylated in *Listeria monocytogenes* teichoic acids. *L. monocytogenes* phage A118 could not infect cells with a GlcNAc component of teichoic acid that was blocked by lectin, but unlike glucosamine, pure GlcNAc did not interfere with A118 infection [51]. The authors explained that the failure of GlcNAc to inhibit phage infection may be due to the unsuitable molecular conformation or charge under the experimental condition [51]. In contrast to our initial expectations, pure GalNAc could not inhibit phage infection, while CPS of wild type could. As phage infection was also inhibited by pure glucosamine and galactosamine, it could be assumed that GalNAc incorporated in CPS would share structural similarity with these pure monosaccharides.

Host range of CPS1 phage was narrow based on the analysis with limited number of *C. perfringens* strains (Appendix A). However, a larger number of strains that include strains closely related with *C. perfringens* ATCC 13124 would be necessary because *C. perfringens* represents quite a diverse group of strains for which only 12.6% of their genome represents core genes shared between strains [52]. The narrow host range of CPS1 and the rapid emergence of CPS1-resistance in *C. perfringens* are two major limitations in practical application of phage CPS1. Further studies on the preparation of the appropriate phage cocktails with different phages or on modification of the CPS1 tail to broaden its host range are required to address these issues.

## 5. Conclusions

In summary, for the first time, we isolated *C. perfringens* virulent phage CPS1 and constructed the EZ-Tn5 random transposon mutant library of *C. perfringens* to identify the phage receptor. In this study, we revealed that CPS served as a receptor for the virulent phage that infects *C. perfringens*. Random mutant library construction using sfDNA in *C. perfringens*, as established in this study, can be used to identify a receptor of other *C. perfringens* phages.

## Figures and Tables

**Figure 1 viruses-11-01002-f001:**
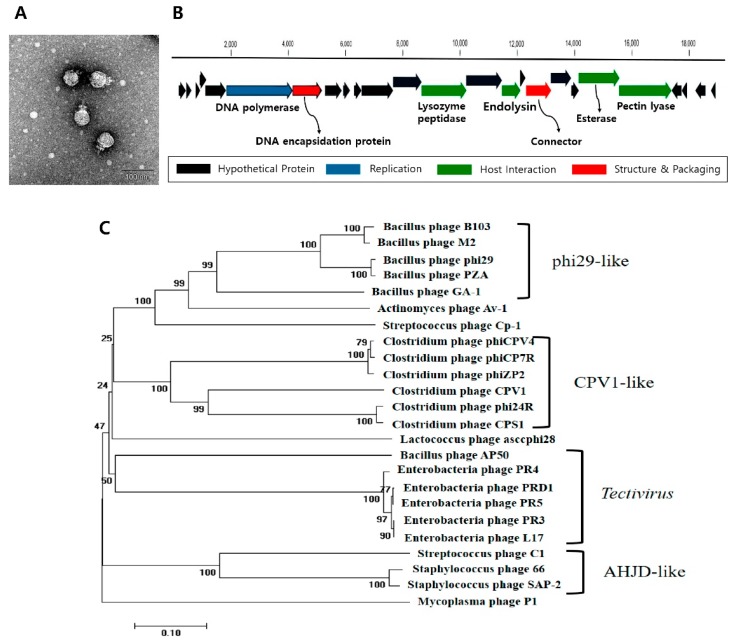
*C. perfringens* bacteriophage capsular polysaccharides (CPS)1 has *Picovirinae* characteristics. (**A**) TEM image of *C. perfringens* phage CPS1. The scale bar indicates 100 nm. (**B**) Genome map of CPS1. Red: structure and packaging, blue: DNA manipulation, green: host lysis, black: hypothetical protein. Twenty-five putative open reading frames (ORFs) were predicted in the CPS1 genome. (**C**) Phylogenetic tree showing the relationship between CPS1 and other *Picovirinae* phages based on type-B DNA polymerase comparisons.

**Figure 2 viruses-11-01002-f002:**
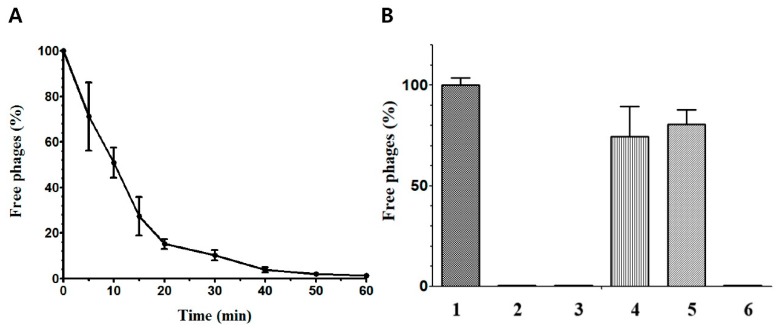
CPS1 binds to polysaccharides on the cell surface. (**A**) CPS1 adsorption assay with ATCC 13124. CPS1 was infected at 30 °C. It was found that 95% of CPS1 adsorbed in 40 min. (**B**) CPS1 adsorption assay with various treatment. 1: BHI (No cells), 2: untreated cells, 3: acetate treatment, 4: 10 mM periodate treatment, 5: 100 mM periodate treatment, 6: proteinase K treatment.

**Figure 3 viruses-11-01002-f003:**
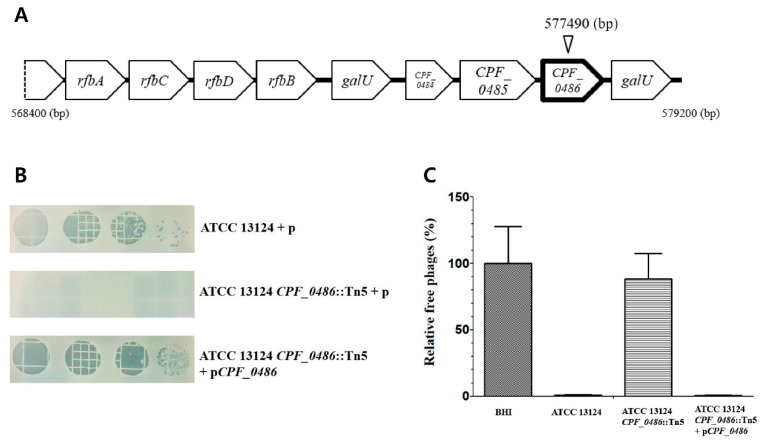
The *CPF_0486* gene was involved in CPS1 resistance. (**A**) Gene order surrounding the *CPF_0486* gene in *C. perfringens* ATCC 13124. The inverted triangle is the EZ-Tn5 transposon insert site. (**B**) Bacteriophage serial diluted dotting assay with ATCC 13124 plus pJIR750, ATCC 13124 *CPF_0486*::Tn5 plus pJIR750, complementary strain. (**C**) CPS1 was not able to attach to the *CPF_0486*::Tn5 mutant. All adsorption assays were performed at 30 °C for 40 min. The means and standard deviations from three independent experiments are indicated by error bars.

**Figure 4 viruses-11-01002-f004:**
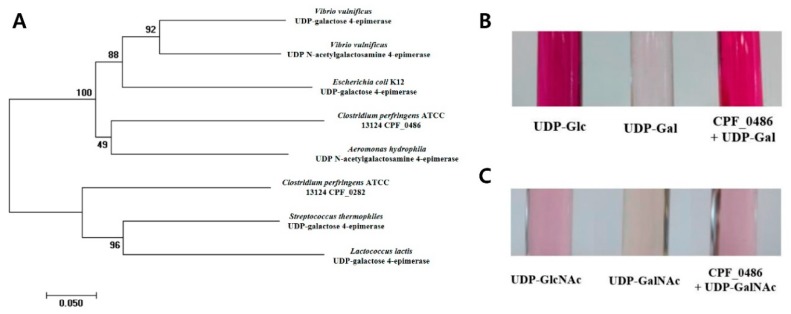
CPF_0486 function was characterized using a colorimetric assay. (**A**) Phylogenetic tree showing the amino acid sequence relatedness of *C. perfringens* CPF_0486 with UDP-glucose 4-epimerase and UDP-*N*-acetylglucosamine 4-epimerase of other species using the MEGA7 software. (**B**) Glucose oxidation (GO) assay. UDP-Glc, UDP-glucose (2 mM); UDP-Gal, UDP-galactose (2 mM); CPF_0486 +UDP-Gal, CPF_0486 with UDP-galactose (2 mM). (**C**) Morgan–Elson reaction assay. UDP-GlcNAc, UDP-*N*-acetylglucosamine (0.2 mM); UDP-GalNAc, UDP-*N*-acetylgalactosamine (0.2 mM); CPF_0486+UDP-GalNAc, CPE_0486 with UDP-*N*-acetylgalactosamine (0.2 mM). Representative colorimetric result from two independent experiments was presented.

**Figure 5 viruses-11-01002-f005:**
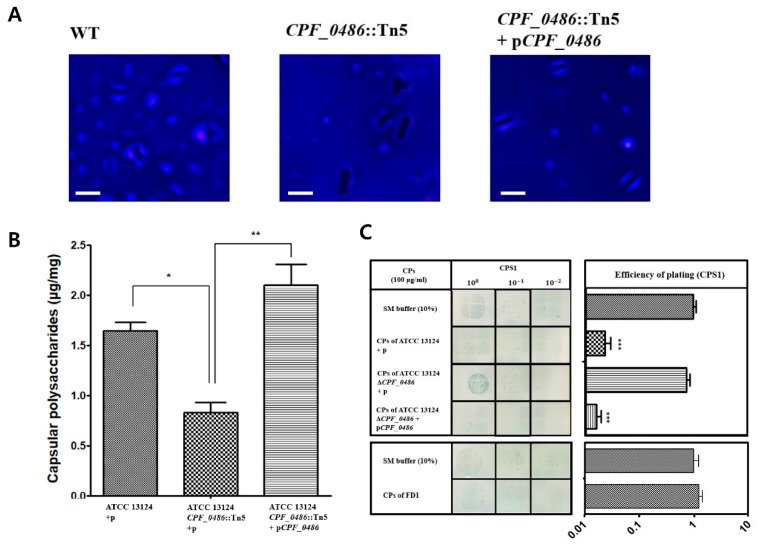
Capsular polysaccharides (CPS) are receptors of CPS1. (**A**) Capsule staining of wild type, *CPF_0486*::Tn5 mutant and complementary strain. The scale bars represent 1 μm. (**B**) Quantification of CPS using a modified phenol sulfuric acid assay. (**C**) Measuring efficiency of plating (E.O.P.) of CPS1 which was preincubated with CPS. Top: CPS of *C. perfringens* ATCC 13124; bottom: CPS of *C. perfringens* FD1. The means with standard deviations (**A**–**C**) of triplicate experiments are shown. * *P* < 0.05, ** *P* < 0.01, *** *P* < 0.0001.

**Figure 6 viruses-11-01002-f006:**
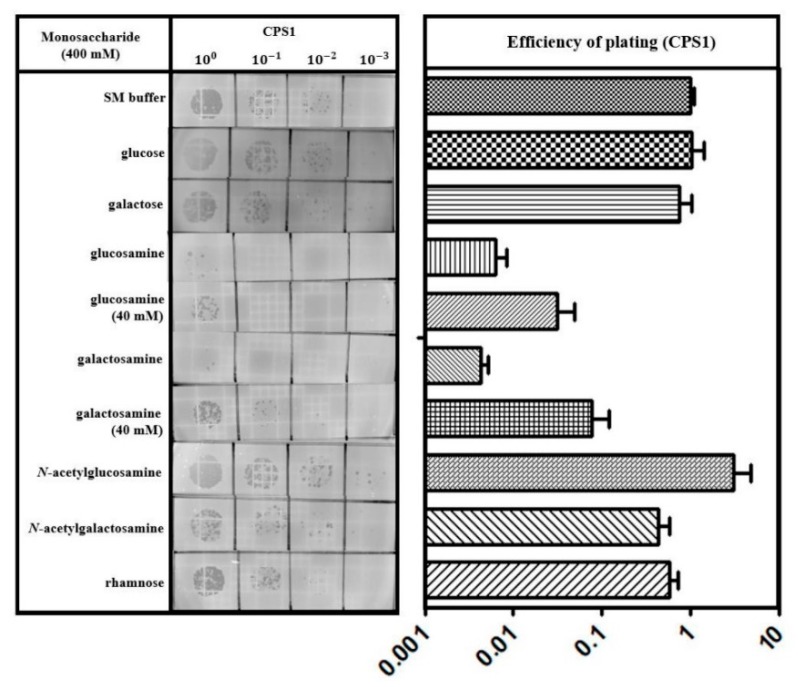
CPS1 adsorption was inhibited by glucosamine and galactosamine. Measurement of the E.O.P. of CPS1, which was preincubated with the indicated monosaccharides. Monosaccharides were used at 400 mM unless otherwise indicated. The results are expressed as the means and standard deviations of triplicate assays.

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
