# Peer review of "Capsular Polysaccharide Is a Receptor of a Clostridium perfringens Bacteriophage CPS1"

_viruses, 2019, doi:10.3390/v11111002_

Round 1
Reviewer 1 Report
This impressive manuscript describes the isolation and characterization of a lytic phage designated CPS1 targeting the environmental pathogen C. perfringens. The work succeeds in identifying the receptor for the phage as the capsular polysaccharide of C. perfringens. The latter is a an unusual receptor for phages targeting gram-positive organisms; it was identified through an optimized transposon mutagenesis method which led to phage-resistant mutants lacking the usual polysaccharide structure, due to mutations in a gene shown to have UDP-Gucose 4-epimerase activity( GalE) as well as UDP-N-acetyl Glucosamine 4-epimerase activity. Interestingly, another mutation in a glycosyl transferase was also identified, but not further characterized in this work.
While the science is rigorously done, my only question, not actually discussed beyond the introduction, is how this work can be applied practically. Two limitations for therapeutic application of this phage cited in the manuscript are : 1)its narrow host range even among C. perfringens strains; and 2) the finding that phage-resistant isolates arise quite readily, and are viable. These could be addressed by using cocktails of similar phages or modifying the phage tail to broaden its host/target range or finding other phages with receptors that are essential for host survival.
Minor points noted that could be corrected:
Pg 4, line 82 –use “appropriate selective antibiotics” , not “proper antibiotics”
Pg 6, line 232 –reference 38 should be [38], not superscript 38.
Pg 8. Line 291 – define GO as Glucose Oxidation assay
Pg 9, line 310 – define term CP used later in figure – specify as “Capsular Polysaccharides (CP)”
Pg 11 line 366 – “glycosyl transferase or galE ” -> the “or” should be “and”
Line 371 –change “that” to ”those”
Author Response
While the science is rigorously done, my only question, not actually discussed beyond the introduction, is how this work can be applied practically. Two limitations for therapeutic application of this phage cited in the manuscript are : 1)its narrow host range even among C. perfringens strains; and 2) the finding that phage-resistant isolates arise quite readily, and are viable. These could be addressed by using cocktails of similar phages or modifying the phage tail to broaden its host/target range or finding other phages with receptors that are essential for host survival.
As pointed out by the reviewer 1, we have briefly discussed the limitations for therapeutic application of phage CPS1 and mentioned the necessity of further study in the discussion section of the revised manuscript (lines 410-413) as follows:
“The narrow host range of CPS1 and the rapid emergence of CPS1-resistance in C. perfringens are two major limitations in practical application of phage CPS1. Further studies on the preparation of the appropriate phage cocktails with different phages or on modification of the CPS1 tail to broaden its host range are required to address these issues.”
Minor points noted that could be corrected:
Pg 4, line 182 –use “appropriate selective antibiotics” , not “proper antibiotics”
As suggested by the reviewer 1, we have changed “proper antibiotics” to “appropriate selective antibiotics” in the revised manuscript (line 182).
Pg 6, line 232 –reference 38 should be [38], not superscript 38.
As pointed out by the reviewer 1, we have corrected the format for reference in the revised manuscript (line 232).
Pg 8. Line 291 – define GO as Glucose Oxidation assay
As indicated by the reviewer 1, we have defined the term “GO” as “Glucose Oxidation (GO) assay” in the revised manuscript (line 294).
Pg 9, line 310 – define term CP used later in figure – specify as “Capsular Polysaccharides (CP)”
As pointed out by the reviewer 1 and 2, we have specified the term “CPS” as the abbreviation of “Capsular polysaccharides” in the revised manuscript (line 313).
Pg 11 line 366 – “glycosyl transferase or galE ” -> the “or” should be “and”
As indicated by the reviewer 1, we have changed the word “or” to “and” in the revised manuscript (line 371).
Line 371 –change “that” to ”those”
As indicated by the reviewer 1, we have changed the word “that” to “those” in the revised manuscript (line 376).
Reviewer 2 Report
The authors of this study have isolated a phage that infects Clostridium perfringens ATCC13124, which they have named CPS1. They have characterized this phage as a member of the order Caudovirales, family Podoviridae, subfamily picovirinae based on the icosahedral head with short non-contractile tail, and small (~19 kb) linear dsDNA genome, which they have sequenced and annotated. Using a transposon library that they generated, they identified a number of C. perfringens genes whose disruption led to inability of the phage to plaque – the majority of these were in rRNA genes, but two were in sugar biosynthesis-related genes. Of the two, they decided to further study the CPF_0486 gene, which was annotated as a putative 4-epimerase, by overexpressing and purifying the encoded enzyme and performed enzyme assays that indicated that it can convert the hexose and HexNAc. Based on a crystal violet-based assay, they observed a halo around the wild-type ATCC13124 strain that was significantly reduced or absent in the CPF_0486 mutant, but restored in the complemented strain, which was attributed to be capsular polysaccharide.
Overall, this is an interesting study that represents a valuable contribution to the field of Gram-positive phage research, and the discovery of a Gram positive phage that uses CPS as a receptor is less commonly described. There are a few issues that should be addressed:
1) Phage host range: It is unclear why the strains that were tested for infection potential were the ones listed? As C. perfringens represents quite a diverse group of strains for which only 12.6% of their genome represents core genes shared between strains, in order to make any claims of host range this phage should be tested against more closely related strains. Please refer to the dendogram at the NCBI page for C. perfringens https://www.ncbi.nlm.nih.gov/genome/?term=clostridium+perfringens (13124 is in yellow when enlarged). Alternatively, the authors can highlight the fact that there is such great diversity amongst C. perfringens strains and that to properly evaluate the host specificity of this phage, a larger sample of strains that includes much more closely related strains would be necessary.
2) Did the authors try enzymatic reactions with UDP-Glc and UDP-GlcNAc? Nucleotide sugar epimerases typically catalyze reversible reactions – in other words, the CPF_0486 enzyme would likely also catalyze UDP-Glc to UDP-Gal and UDP-GlcNAc to UDP-GalNAc. Given that UDP-Glc and UDP-GlcNAc are common precursors in the bacterial cell and their galacto configuration counterparts are not, it is likely that the role of this enzyme in the cell is to generate either UDP-Gal or UDP-GalNAc, both of which are found in the CPS of ATCC13124. The ability or inability of the enzyme to catalyze UDP-GlcNAc to UDP-GalNAc would at least indicate whether it is possible the enzyme could be responsible for the GalNAc residue, as suggested by the authors in the discussion.
While it would be helpful for the authors to perform the enzyme assay with CPF_0486 and UDP-GlcNAc to help clarify which potential role(s) the enzyme could have in CPS biosynthesis, at least the activities demonstrated in the paper are consistent with the possibility of generating the Gal and GalNAc residues in the CPS. However, the authors should at least describe that the UDP-GlcNAc to UDP-GalNAc role is a possibility and that this reaction would need to be performed to confirm that possibility, as well as the possibility that this enzyme is the UDP-Glc to UDP-Gal or UDP-GlcNAc to UDP-GalNAc 4-epimerase, there is a good possibility that the other enzyme, which may be CPF_0282, would be able to partially compensate for the loss of CPF_0486, resulting in low levels of CPS in the mutant rather than complete loss.
3) While the authors provide evidence that CPS acts as a receptor for this phage, I think it is premature/overstating that CPS is THE receptor. Several phage are known to have more than one receptor and, although the author’s explanation of why glucosamine and galactosamine affected EOI but GlcNAc and GalNAc did not is acceptable, this result and the inability to completely inhibit infection with the CPS leaves open the possibility that there may also be a second receptor (possibly even a yet unidentified glycan with different composition). As a result, the authors should change their wording from “the receptor” to “a receptor”
4) The electroporation section is confusing and is missing information such as cuvette width and electroporator settings such as kV, time constant [if used], and capacitance [if known] (or listing the electroporator instrument and which pre-setting was used if these settings are not listed in the instruction manual). Suggestions for re-wording are listed in the editorial section below
5) While it is likely that the method used to purify CPS yielded CPS vs LTA, WTA, or glycans from glycosylated proteins (if they exist in C. perfringens), ideally the authors should demonstrate this in some way. While MS or NMR would be ideal, a simple alternative would be to run an SDS-PAGE with the sample and visualize by silver-staining, which would at least demonstrate that the size/migration is consistent with CPS.
6) Figure sizing: As currently sized, the text or images in several figures are too small to clearly read or see the result. In addition to re-sizing (which may be an editorial issue rather than an author issue).
Font too small: Figs 1, panel B; Fig 2, panel A; Fig 3, panel A
Figure too small to see result: Figure 5, panel A. The images are far too low for such a key result in support of the authors’ claim of CPS being a receptor for this phage
Do the authors think Fig 1, panel C is necessary? It is possible to state that there is a 166-bp inverted terminal repeat sequence and refer to the accession number.
Editorial revisions
Throughout
* UDP-NAcGal and UDP-NAcGlc are incorrect. Change to UDP-GalNAc and UDP-GlcNAc
* CPs is incorrect, and should be changed to CPS. The acronym CPS is standard in glycobiology and is used for both singular and plural forms of capsular polysaccharide.
* Do not italicize CPF_0486 when referring to the protein/gene product, only when referring to the gene.
* His-tag nomenclature: Both His6 and His6 are found in the text. For consistency, use only one of these.
P1, line 14: if saying virulent, must state host (eg. Here we isolated a virulent C. perfringens phage,…”
P2, line 78: “incubating”, not “reacting”
P2, line 92: “…of CPS1 and the ATCC 13124…”
P3, line 100-112. The electroporation/mutagenesis description section is somewhat confusing, and needs to be reworded. Here is a suggested rewording/reorganization:
“2.5 molecular genetic methods
Since C. perfringens ATCC13124 is extremely difficult to modify using conventional electroporation methods, we optimized electroporation conditions by including a short non-coding DNA fragment (sfDNA, 300 bp) in the transformation mixture and electroporated at 30oC, using plasmid pJIR750 as control DNA (include cuvette width, kV, time constant, and capacitance [if known] here). These optimized transformation conditions resulted in 10-fold higher efficiency compared to electroporation at RT without the sfDNA.
The C. perfringens ATCC13124 cpf_0486::Tn5 mutant was isolated from an EZ-Tn5 mutagenesis library, as follows: A random C. perfringens ATCC 13124 mutant pool was generated using the EZ-Tn5 kit.”, or similar.
P3, line 106. Please fill in the blank with the appropriate descriptor: “The EZ-Tn5 transposon ___ in composed of…” Is it the kit? Or the reaction?
P3, line 107. Use “Erm” or “Em”, not “erm”
P3, line 111. Replace “…using primers by Microgen Inc (Table 2S)” with “…using the appropriate primers listed in Table 2S.”
P3, line 121. “…, and purified using the …”
P3, line 122. “The constructed N-terminally His6-tagged CPF_0486 expression vector was transformed…”
P3, line 124. “The gene encoding CPF_0486 with an N-terminal His6-tag was expressed…”
P3, line 135. Does the storage buffer contain no Tris or other buffering agent? Or is the storage buffer include 4 mM NAD+ and 50% glycerol?
P3, line 139. Add“…with the colorimetric assay of Fry et al. [31]”
P3, line 143. “4-epimerase activity with glucose was assayed by …”
P4, line 157. Were the cells diluted or pelleted and resuspended? What media or buffer was used to dilute or resuspend the cells?
P4, line 192. Please define D.W. If meaning deionized water, the standard short form is DI water.
P6, lines 260-262. Reorganize these sentences like the following or similar: “We chose…considering that this gene is in a polysaccharide biosynthesis cluster (Fig 3A) [41] and the periodate…”
P9, line 312: “Quantitation of CPS using a modified phenol sulfuric acid assay”
Fig 6. Include a statement in the legend to the effect of “Monosaccharides were used at __ mM unless otherwise indicated”
P10, 345. “…belonging to the subfamily Picovirinae [38], although only 3 such phages have been identified to date”
P10, line 357. “…of a phage infecting Gram positive bacteria via CPS as a receptor.”
367: re-word, since CPF_0496 was not characterized prior to this study, but a homologue of it was, while family 2 glycosyltransferases are a broad group of enzymes that catalyze a wide array of sugars
